An evaluation of alternative methods for constructing phylogenies from whole genome sequence data: a case study with Salmonella

Pettengill James B.
Luo Yan
Davis Steven
Chen Yi
Gonzalez-Escalona Narjol
Ottesen Andrea
Rand Hugh
Allard Marc W.
Strain Errol Errol.Strain@fda.hhs.gov
Center for Food Safety & Applied Nutrition, U.S. Food & Drug Administration , College Park, MD , USA
Crandall Keith
Electronic publication date: 2014 Oct 14
Publication date: 2014
Volume: 2
Electronic Location ID: e620
Received 2014 Apr 16; Accepted 2014 Sep 23
Copyright: © 2014 Pettengill et al.
Copyright year: 2014
Copyright holder: Pettengill et al.
License: This is an open access article distributed under the terms of the Creative Commons Attribution License, which permits unrestricted use, distribution, reproduction and adaptation in any medium and for any purpose provided that it is properly attributed. For attribution, the original author(s), title, publication source (PeerJ) and either DOI or URL of the article must be cited.
License URL: https://creativecommons.org/licenses/by/4.0/

Keywords: Salmonella, Outbreak, Congruence, Phylogenetics, Next generation sequencing, Single nucleotide polymorphism

Funding: This work was supported by the Center for Food Safety and Applied Nutrition at the US Food and Drug Administration. The funders had no role in study design, data collection and analysis, decision to publish, or preparation of the manuscript.

==============================
Comparative genomics based on whole genome sequencing (WGS) is increasingly being applied to investigate questions within evolutionary and molecular biology, as well as questions concerning public health (e.g., pathogen outbreaks). Given the impact that conclusions derived from such analyses may have, we have evaluated the robustness of clustering individuals based on WGS data to three key factors: (1) next-generation sequencing (NGS) platform (HiSeq, MiSeq, IonTorrent, 454, and SOLiD), (2) algorithms used to construct a SNP (single nucleotide polymorphism) matrix (reference-based and reference-free), and (3) phylogenetic inference method (FastTreeMP, GARLI, and RAxML). We carried out these analyses on 194 whole genome sequences representing 107 unique Salmonella enterica subsp. enterica ser. Montevideo strains. Reference-based approaches for identifying SNPs produced trees that were significantly more similar to one another than those produced under the reference-free approach. Topologies inferred using a core matrix (i.e., no missing data) were significantly more discordant than those inferred using a non-core matrix that allows for some missing data. However, allowing for too much missing data likely results in a high false discovery rate of SNPs. When analyzing the same SNP matrix, we observed that the more thorough inference methods implemented in GARLI and RAxML produced more similar topologies than FastTreeMP. Our results also confirm that reproducibility varies among NGS platforms where the MiSeq had the lowest number of pairwise differences among replicate runs. Our investigation into the robustness of clustering patterns illustrates the importance of carefully considering how data from different platforms are combined and analyzed. We found clear differences in the topologies inferred, and certain methods performed significantly better than others for discriminating between the highly clonal organisms investigated here. The methods supported by our results represent a preliminary set of guidelines and a step towards developing validated standards for clustering based on whole genome sequence data.

Background

The increasing availability and decreasing costs of next-generation sequencing (NGS) has made it possible to apply this technique to investigate non-model organisms and myriad questions within evolutionary and molecular biology. Next-generation sequencing as a means to acquire whole genome sequences (WGS) has also revolutionized applied research, where it is now, for example, being employed to investigate issues related to public health such as pathogenic outbreaks (Lienau et al., 2011; Parkhill & Wren, 2011; Roetzer et al., 2013; Underwood et al., 2013). In this situation, the vastly increased information content in WGS data provides a genome perspective of differences among isolates, which can be used to identify gene specific differences associated with virulence and pathogenicity. The information content within WGS data also provides superior discriminatory power with which the evolutionary relationships among highly clonal strains can be inferred. This phylogeny can then be used to identify clinical cases properly belonging to outbreaks and establish links between clinical and environmental isolates.

Outbreaks have been studied with NGS both retrospectively, as a means to better understand a past outbreak, and during an outbreak, to aid in real time decision-making. A well known example illustrating the utility of NGS to investigate a food-borne disease outbreak is that of the virulent Shiga toxin (Stx)-producing Escherichia coli O104:H4 associated with contaminated sprouts in Germany; NGS data has been used to better understand differences in virulence (Bielaszewska et al., 2011) and evolutionary relationships (Mellmann et al., 2011) among outbreak and non-outbreak isolates. A number of recent cases of salmonellosis also have been investigated by clustering isolates based on SNP (single nucleotide polymorphism) differences identified by WGS data. For example, two studies, which used data produced on different NGS platforms and using different bioinformatic methods, unequivocally identified the source of a multistate outbreak of Salmonella enterica subsp. enterica serovar Montevideo associated with spiced-meat (Allard et al., 2012; den Bakker et al., 2011). With regards to human diseases, phylogenetic reconstruction based on WGS data proved useful in the investigation and determination of the source of Vibrio cholerae associated with an outbreak that occurred in Haiti in 2010 (Chin et al., 2011; Hendriksen et al., 2011; Keim et al., 2011).

Recent studies in the hospital setting illustrate how NGS can contribute to the real-time management of disease outbreaks. Koser et al. (2012) utilized WGS data to rapidly investigate an outbreak of methicillin-resistant Staphylococcus aureus (MRSA) within a neonatal intensive care unit. Such an approach was able to distinguish among isolates that belonged to a single lineage, which was likely impossible using traditional typing methods. As a result, NGS data was shown to rapidly (i.e., within a timeframe that could influence patient health) provide clinically important data (Koser et al., 2012). Health officials investigating a hospital outbreak of Klebsiella pneumoniae found that using whole-genome sequencing yielded “actionable insights” by discerning among alternative transmission routes and, thus, containment of the infection (Snitkin et al., 2012).

The recent publication dates of the studies discussed above illustrate just how new the use of NGS data within public health is. Given the potential impact of public health decisions based on NGS data, the robustness of NGS-based results must be evaluated. Three factors that may differ within or among studies are (1) the NGS platform on which the sequences were generated, (2) the method used to detect variant sites and create a SNP matrix, and (3) the phylogenetic inference method used to cluster samples. With regards to NGS platform, it is well documented that performance (e.g., error rates and error structure) differs among them (Harismendy et al., 2009; Loman et al., 2012; Mardis, 2013; Shendure & Ji, 2008). Although some may argue this is not a significant issue as certain platforms are no longer maintained or less likely to be used in the future (e.g., SOLiD and Roche 454), the fact that large amounts of data have been produced under such platforms means that NGS platform artifacts will need to be accounted for within analyses that incorporate historical data. The rapid incorporation of NGS data within many disciplines has produced many bioinformatic tools, all of which may produce different results when trying to accomplish the same task. For example, comparative studies have shown that not all de novo assemblers are equal (Bradnam et al., 2013; Magoc et al., 2013; Salzberg et al., 2012; Zhang et al., 2011) nor are mapping algorithms (Hatem et al., 2013; Ruffalo, LaFramboise & Koyuturk, 2011) or variant detection algorithms (Cheng, Teo & Ong, 2014). As for phylogenetic inference, performance comparisons of recently developed maximum-likelihood methods to handle the large matrices often associated with WGS data show that those methods do not always produce the same topology (Liu, Linder & Warnow, 2011).

In this study, our objective was to advance the use of WGS data within the public health realm, and in general, by investigating the robustness of the inferred phylogenetic relationships to several key factors that can vary among analyses. Here, we define robustness as the sensitivity of results to differences in the acquisition and analysis of data. In our case, data acquisition and analysis represent the combination of different sequencing platforms, SNP detection methods, and phylogenetic inference algorithms; the results are the inferred topology. We first present the reference-based approach we have developed to identify SNPs among the WGS data we are routinely producing. We then quantified the congruence among our method and three other SNP detection methods (one reference-based and two reference-free) by comparing the topologies produced with the same phylogenetic inference package. We evaluated the influence of sequencing platform by determining the degree to which replicate runs of the same strain clustered together and the number of differences among replicate runs of the same strain on the same platform. We also assessed the influence of three different phylogenetic inference methods by comparing topologies created using the same data matrix. These analyses were performed on a set of Salmonella enterica subsp. enterica Serovar Montevideo isolates sequenced on five different NGS platforms. Many of the isolates were implicated in an outbreak of salmonellosis and the evolutionary relationships of a subset of isolates were investigated within two previous studies (Allard et al., 2012; den Bakker et al., 2011).

Materials and Methods

Sequence data

All 194 samples included in this study were downloaded from the SRA archive (Table S1) from which fastqs or sff files were extracted using the SRA Toolkit v2.1.6. SOLiD SRA files were converted to colorspace fastq (csfastq). The 194 samples represent 107 unique strains, 3 of which are Salmonella enterica ssp. enterica serovar Pomona that served as an outgroup and the others are Montevideo (Table 1). Samples were from one of five platforms: Illumina MiSeq and HiSeq, Life Technologies IonTorrent, Roche 454 FLX Titanium, and Applied Biosystems SOLiD. Thirty-two strains were sequenced on more than one sequencing technology (Table 1 and Table S1). All but the 454 runs for strain IA_2009159199 are replicates from independent passages (overnight cultures) and, thus, from different library preparations for sequencing.

Table 1 Distribution of samples by platform.

The number of samples run on both platforms; diagonal elements contain the number of sequencing runs per platform.

	454	HiSeq	IonTorrent	MiSeq	Solid	
454	68	0	17	17	0	
HiSeq		1	0	0	0	
IonTorrent			30	24	6	
MiSeq				47	5	
Solid					48	

In addition to the full dataset of 194 samples, we constructed a dataset containing only those samples present in Allard et al. (2012). With the Allard dataset (116 runs, 47 strains) our primary focus was on how the conclusions within a traceback investigation (i.e., linking clinical isolates that sickened people to environmental sources) would differ under the three factors being considered (i.e., NGS platform, SNP detection method, and phylogenetic inference method). The conclusions based on NGS data within that study are also supported by epidemiological data.

Given that we analyzed empirical data, we do not know the true evolutionary relationships among the individuals. However, we do have replication of a number of isolates and, therefore, have expectations for how those isolates should cluster (i.e., be monophyletic) and the number of SNP differences that should exist among them (i.e., zero). For analyses comparing topologies (see below), we were not able to determine the accuracy of the methods but rather focused on the variance among methods where we were working under the assumption that methods that produced more similar trees are superior to methods that result in different topologies.

Variant detection methods

We investigated the performance of four different SNP detection methods, which can be broadly separated into reference-based and reference-free. The primary difference between the two types is that reference-based methods use a genome, either a closed or draft assembly, to which reads are mapped and variant positions are called. In contrast, the reference-free approach determines variant sites in the absence of a reference genome by comparing portions (e.g., k-mers) among all samples.

Reference-based methods

For the reference-based methods we used the closed genome of strain 507440-20 that was sequenced on the PacBioRS II (Pacific Biosciences, Menlo Park, CA, USA) and assembled using SMRT Analysis v2.0.1 (Genbank ID CP007530.1). We chose this as the reference genome as it represented the closest Montevideo environmental strain to the outbreak (Allard et al., 2012). Raw reads from each sample were mapped to the reference genome using default settings within Bowtie2 v2.1.0 (Fig. 1) (Langmead & Salzberg, 2012). SOLiD data was mapped using Bowtie v1.0.0 (Langmead et al., 2009), which has the ability to handle colorspace data. There are many other mappers that could be used but we chose not to evaluate mappers as their performance has been evaluated elsewhere (Hatem et al., 2013; Ruffalo, LaFramboise & Koyuturk, 2011). We then used samtools v0.1.18 (Li et al., 2009) to sort the BAM file from Bowtie and produce a pileup file for each sample.

Figure 1 Workflow describing the analyses conducted in this study.

Dotted boxes highlight the four effects (sequencing platform, SNP detection method, and phylogenetic inference) that are accounted for at that step in the process. The platforms and methods we currently use in our investigations are in bold italic font.

Under our method, the resulting pileup files from samtools are processed using VarScan2 v2.3 (Koboldt et al., 2012) to identify high quality variant sites using the mpileup2snp option, which are stored in individual .vcf files. We used the default values for variant detection except the minimum variant allele frequency was set to 0.90. As a result, we refer to this method as VarScan. We used a custom python script to parse the .vcf files and construct a SNP matrix, which included two steps. First, a ‘snplist’ file is generated by identifying all variant sites across the individual .vcf files, which represents all the positions that will make up the SNP matrix. Second, for each sample an additional pileup file is generated that contains only those positions found in the ‘snplist’ file. It is from these second pileup files that we determine the nucleotide state for each position in SNP matrix. Determining the state of each position in the snplist file is based on the following rules: (a) if different nucleotides were called at the position, the one with frequency larger than 50% was the consensus call for that position but this threshold can be altered to match a different user defined value; and (b) if different nucleotides were called at a position but none had a frequency larger than 50%, that position for that individual was coded as missing data. Positions identified as indels in the .vcf file were ignored in the construction of the SNP matrix. Additional information (e.g., our code and instructions) is at https://github.com/CFSAN-Biostatistics/snp-pipeline.

To evaluate the sensitivity of our reference-based method, we also constructed SNP matrices using the variant detection algorithm implemented in SolSNP v1.11 (Fig. 1) (Robbins et al., 2011). Like our own method, reads were mapped to a closed reference using Bowtie1 or Bowtie2. However, in contrast to our method that uses VarScan, the resulting bam files were then processed with SolSNP, which uses a Kolmogorov–Smirnov statistic as a distance measure to call the most likely nucleotide state with respect to the reference. The resulting .vcf files were parsed using our python script to construct the SNP matrices for downstream phylogenetic analyses. All SolSNP analyses were conducted using the default parameters.

For the reference-based methods, in addition to the all matrix that consists of all SNPs discovered in the analysis that had 0.7%–0.8% missing data (Table 2), we constructed a core matrix that consisted of only those SNP positions for which every sample had a nucleotide called (Fig. 1).

Table 2 SNP matrix characteristics.

Summary of the differences across the SNP detection methods by matrix (Core, All, 50%) in terms of the number of SNPs, percent missing data, and number of identical sequences.

SNP detection	Method	Matrix	Number of SNPs	Missing	N identical b	
Reference-based	VarScan	Core	8,056	na	114	
		All	49,307	0.7%	1	
	SolSNP	Core	7,179	na	123	
		All	45,388	0.8%	2	
Reference-free	kSNP de novo	Corea	0	na	na	
		50%	51,261	11%	0	
		All	73,236	37%	0	
	kSNP raw	Core	749	na	135	
		50%	54,157	11%	1	
		Alla	2,990,475	96%	0	
Notes.

na, not applicable.

a Not analyzed.

b Number of identical sequences within the matrix.

Reference-free methods

We used the program kSNP v2.0 (Gardner & Hall, 2013) to produce SNP matrices without using a reference sequence. kSNP uses a k-mer approach to identify homologous single nucleotide polymorphisms among a group of individuals. Briefly, the program uses jellyfish (Marcais & Kingsford, 2011) to index all draft genomes into k-mers and SNPs are identified using MUMmer (Kurtz et al., 2004). Although we are aware of at least one other k-mer based approach, it requires paired-end data (Schwartz et al., 2013).

We used kSNP on two different treatments of the sequence data. Under the first we used filtered fastq files; reads were filtered using seqtk (https://github.com/lh3/seqtk), which masked bases with Q-scores <20. Under the second treatment, we performed de novo assemblies that were then used as the input for kSNP. We chose an assembler that was well suited for the sequencing platform from which the reads came, which was CLC Genomics Workbench 6.0.5 (CLC Bio, Cambridge, MA, USA) for the Illumina, SOLiD, and IonTorrent data and Newbler v2.6 for the 454 data. As was the case with the mapping software under the reference-based approach, our objective here was not to evaluate multiple de novo assemblers as that has been done elsewhere (Magoc et al., 2013; Salzberg et al., 2012; Zhang et al., 2011). In both kSNP analyses, the k-mer size was set to 25.

For both the de novo and raw read approaches, three SNP matrices were produced, an “all” matrix, a “majority” matrix, and a “core” matrix. The all and core matrices are as defined for the reference-based methods. The majority matrix contained only those positions in the all matrix for which at least 50% of samples had a nucleotide base called. With the de novo approach, the core matrix had 0 nucleotides present (likely due to missing data associated with the poorer assemblies; Fig. 2) so we only analyzed the all and majority matrix under that approach. The all matrix constructed with the raw reads had 96% missing data and contained 2,990,475 positions (Table 2), which does not seem biologically reasonable as it suggests that nearly 66% of the genome contains variable positions. As a result, we do not perform any of the subsequent analyses on this matrix.

Figure 2 Characteristics of sample reads by platform.

(A) Average number of reads produced, (B) percent of reads mapped to the reference sample, (C) the number of contigs within the assembly, (D) total number of basepairs assembled; dashed red line represent the bps in the reference, and (E) the N50 of the assembly. Boxes depict the interquartile (IQR) range and whiskers indicate 1.5 IQR; the horizontal black line represents the mean. The IonTorrent platform is abbreviated as IonT. Points represent observed values; points were also offset from one another (‘jittered’) to reduce overlap. See Table 1 for sample sizes per NGS platform.

Analyses of replicate runs

Within each matrix, we calculated the proportional pairwise number of SNP differences among replicates of a strain run on the same NGS platform. Analyses were conducted using the dist.dna function within the ape package (Paradis, Claude & Strimmer, 2004) in R (R Development Core Team, 2011)

We used the genealogical sorting index (gsi) (Cummings, Neel & Shaw, 2008) to quantify the degree to which replicate runs of the same strain clustered together, which provides insight into the robustness of the phylogenetic relationships to differences in sequencing platform and bioinformatic approaches. The gsi varies from 0 (a completely random assortment of individuals from the same group (e.g., strain) on the tree) to 1 (replicates are reciprocally monophyletic). Generally speaking, the gsi is based on the number of nodes uniting all individuals from the same group divide by the observed number of nodes uniting those members. We calculated the gsi statistic for two different cases: (1) all replicates of a given strain comprise a single group and (2) replicates run on the same platform comprise a group. We chose these two groupings because they allowed us to characterize, as best we could given the dataset, the influence of within and between platform variation. To account for phylogenetic uncertainty, we estimated the weighted gsi value under which 100 bootstrap replicates for each matrix (e.g., VarScan, kSNP de novo) by phylogenetic inference comparison were analyzed.

Phylogenetic analysis

Topologies based on each of the matrices created under the different SNP detection methods were created using three phylogenetic inference methods (FastTreeMP (Price, Dehal & Arkin, 2010), GARLI (Zwickl, 2006), and RaxML (Stamatakis, 2006)). Each of the three methods uses a different approach to increase computational feasibility when analyzing large matrices. When evaluating the robustness of the results to different SNP detection methods or NGS platforms, we only compared topologies produced using the same phylogenetic inference method. To assess the robustness of results to differences in phylogenetic inference method, we only compared topologies constructed with the same SNP matrix.

For the FastTreeMP v2.1.7 analyses, we used the -gtr -nt -cat 4 flags; analyses were run on a single desktop machine with dual 2.93 GHz 6-Core Intel Xeon processors and 48 GB of shared RAM. We used phylip’s seqboot (Felsenstein, 1989) option to produce 100 bootstrap replicates of the dataset, each of which were then analyzed with FastTreeMP as was done with the non-bootstrapped dataset. Given that the other two methods were carried out conducting traditional non-parametric bootstrapping we thought it appropriate to do the same with FastTreeMP rather than rely on the SH-like local support values that program can also produce. We used the default parameter setting for the GARLI v2.0 analyses including the GTR + I + Γ. We ran 100 replicate GARLI analyses for each of the observed matrices, and present the topology with the best likelihood score to which support values based on 1,000 bootstrap replicates were added. GARLI analyses were performed on a high performance computer within the FDA’s Scientific Computing Lab; batches of up to 20 runs were performed simultaneously on this resource where compute nodes contain 8 Intel Xeon 2.67 GHz processors and 24 GB of shared RAM. For RAxML v7.9.1, we used raxmlHPC-PTHREADS-AVX, the -m GTRCAT flag, and -b flag to perform 1,000 non-parametric standard bootstrap replicates; analyses were run on a Linux machine with 32 Intel Xeon 2.00 GHz processors and 64 GB of shared RAM.

Topological congruence

We quantified the congruence among topologies using the symmetric difference statistic (Steel & Penny, 1993) (i.e., the Robinson–Foulds distance, Robinson & Foulds, 1981) implemented in the phangorn package (Schliep, 2011) in R. The symmetric difference statistic is twice the number of internal nodes that differ in their branching between two trees. Our interpretations of the symmetric difference statistic are based on the assumption that methods or matrix types that result in similar topologies are superior to those that do not. As was the case with the gsi analyses, to account for phylogenetic uncertainty, we calculated the symmetric difference between all pairwise comparisons of 100 bootstrap replicates (e.g., pairwise comparisons between 100 GARLI bootstrap replicates of the VarScan and kSNP de novo matrices).

Results

Mapping and de novo assembly summary statistics

Although the number and lengths of reads varied greatly among the platforms, at least 50% of the reads could be mapped for all but a few samples (Fig. 2). The number of contigs and N50 values varied among platforms but for the majority of the samples, except perhaps the SOLiD runs, the length of the assemblies were close to the expected 4.5 Mbp genome size of an S. Montevideo (Fig. 2). These mapping and assembly summary statistics are promising in that they suggest that similar amounts of data can be extracted from runs on different platforms.

Platform effects

To characterize platform variability, we examined replicate runs of the same sample sequenced on the same platform (Fig. 3). Ideally these replicates would have no proportional pairwise differences and be identical. In reality we found that the 454 platform had the greatest number of differences, followed by Ion Torrent, and then by MiSeq. This pattern was generally consistent across the different variant detection methods except for the kSNP de novo datasets where IonTorrent replicates were more different from one another than on the 454 platform. (The SOLiD platform had no replicate runs and was not assessed.) We note that many of the replicate samples are not pure platform replicates, but were from independent cultures often performed at different labs, which means our observed differences are not just due to platform variability.

Figure 3 Proportional number of SNP differences between replicate runs of the same strain on the same platform across the different matrices investigated.

Boxes depict the interquartile (IQR) range and whiskers indicate 1.5 IQR; the horizontal black line represents the mean.

To examine the effect of platform on clustering, we computed the weigthed gsi with classes assigned by strain and with classes assigned by the combination of strain and platform. Regardless of matrix type and SNP detection method, the strain replicates did not cluster together and, thus, gsi values for many strain classes were <1 (Fig. 4). When classes were defined by the combination of strain and platform, there was a much higher average gsi score. This implies that replicates do cluster together more reliably when run on the same platform. However, despite the differences in reproducibility and gsi values between the classes, we did not find evidence to suggest that there is a strong platform effect when clustering (e.g., clades were not found consisting of only samples run on a single platform; Fig. S1).

Figure 4 Boxplots of gsi values for the different topologies inferred and different groupings.

Boxes depict the interquartile (IQR) range and whiskers indicate 1.5 IQR; the horizontal black line represents the mean. Points represent observed values; points were also offset from one another (‘jittered’) to reduce overlap. See Table S2 for the number of representatives under each grouping.

Matrix size effects

Our results indicate that a matrix that accepts some level of missing data, while not including all sites for which only a few samples have nucleotides states, is optimal for phylogenetic inference. For example, topologies created using core matrices within which there is no missing data had a greater degree of topological incongruence (Fig. 5), greater number of identical sequences (Table 2), and poorer phylogenetic resolution (Fig. 6) when compared to topologies inferred with non-core matrices. However, the all SNP matrix created using the reference-free method with de novo assemblies had approximately 20,000 more SNPs compared to the majority matrices under the reference-free approaches and the all matrices under the reference-based methods, which is an increase in matrix size of approximately 30%. Although some of these additional SNPs may be due to mobile elements/gene presence–absence, it is also likely that many of them are erroneous, which may explain the decrease in bootstrap support despite the larger matrix when compared to the de novo majority matrix (Fig. 6). An additional explanation for the decreased support is that the larger matrices also have more missing data that may increase phylogenetic uncertainty (Roure, Baurain & Philippe, 2013). The higher proportional differences among replicates observed in this matrix also suggests that the additional SNPs may be erroneous and, thus, such matrices should be avoided (Fig. 3). In conclusion, within a reference-based approach, it is probably optimal to use all SNP sites detected rather than cull positions with missing data; for a reference-free approach the threshold of missing data that results in optimal information content is more complicated and depends, in part, on the evolutionary breadth of the samples being investigated.

Figure 5 Violin plots of changes in the symmetric differences statistics across SNP matrices.

Figures show the kernel density estimate of the symmetric differences, broken down by three factors (1. phylogenetic inference package, 2. type of comparison, and 3. type of matrix). Individual observations are omitted due to the large number of them.

Figure 6 Phylograms inferred with RAxML under each of the different matrices created either using a reference-free or reference-based approach.

Red circles represent bootstrap support values greater than 0.85.

Differences due to SNP detection method

The proportion of SNP differences among replicate samples run on the same platform varied, although not substantially, across the matrices created under either the reference-free or reference-based approach (Fig. 3). Given that all methods are (1) working with the same raw reads and (2) either the same assemblies or bam files for kSNP de novo and reference based approaches, respectively, these differences are predominantly due to the differences in algorithms used to identify variant sites.

The method used to construct the SNP matrix had a significant effect on the degree of congruence among trees (Figs. 5 and 7). For example, trees constructed using a reference-based method were more similar to one another (i.e., had a lower symmetric difference score) than those produced under a reference-free approach or when comparing trees created under the different methods.

Figure 7 Violin plots illustrating the differences among inference programs.

Figures show the kernel density estimate of the symmetric differences and topologies are based on the same matrix but different inference methods for both the full 194 sample dataset and for the pared down Allard dataset.

Phylogenetic inference method

For both the reference-free and reference-based approaches there was greater topological congruence when comparing a tree produced under RAxML and GARLI than when either of the trees produced under those methods were compared to one inferred with FastTreeMP. For example, more topological rearrangements are required to reconcile FastTreeMP trees with either GARLI or RAxML (Fig. 7). As we do not know the true topology, we note that when comparing the different inference methods we are working under the assumption that methods that result in similar topologies are optimal. This also means that we cannot assess the accuracy of the methods but, rather, we are estimating the variance among the topological differences they produce.

Allard et al. (2012) samples

To explore the consequences of differences observed among topologies as a result of a combination of NGS platform and SNP detection method within an outbreak scenario, we pared down the dataset to include the 116 runs that represented only the 47 strains present in Allard et al. (2012). Here, we refer to outbreak isolates as those whose phylogenetic placement clusters them with the clinical samples from infected individuals and non-outbreak samples are those whose topological placement is outside of the cluster containing the focal clinical samples. We focused on whether outbreak runs clustered together; if they did, that would provide support that significant patterns related to outbreak investigations are robust to combining runs across different sequencing platforms and using different SNP detection and/or phylogenetic inference packages.

When using RAxML or GARLI to infer a topology, the VarScan, SolSNP, kSNP de novo and kSNP raw matrices constructed with the Allard et al. (2012) dataset resulted in strong support (100 to 81 BP (bootstrap probability); Table 3) differentiating non-outbreak from outbreak isolates that were consistent with the results of that study (Fig. 8). The de novo reference-free approach had the lowest level of bootstrap support among both the GARLI (BP = 81) and RAxML (BP = 91) trees. The results using FastTreeMP were only congruent with the previous results when analyzing the kSNP de novo, kSNP all and VarScan core matrices, the latter two had low support (≤65 BP) (Table 3; Fig. S2). Interestingly, we found that both reference-based all matrices had six fixed SNP differences that differentiated the outbreak and non-outbreak clades; there was only a single fixed SNP difference within the reference-free majority matrices. The difference between the two methods is due to the exclusion under the reference-free approach of SNPs that are within the k-mer distance of one another (i.e., 25 bp).

Figure 8 Topologies inferred with RAxML on matrices that had been pruned to only include strains present in Allard et al. (2012).

Tips are color-coded based on whether the strain was called as part of the outbreak (red circles) or not part of the outbreak (gray circles).

Table 3 Bootstrap support values under each phylogentic inference method and SNP matrix for the bifurcation differentiating outbreak from non-outbreak samples in the Allard et al. (2012) dataset.

Matrix	RAxML	GARLI	FastTree	
VarScan	93	99	na	
VarScan core	na	na	55	
SolSNP	98	97	na	
SolSNP core	na	na	na	
kSNP de novo	91	81	83	
kSNP de novo all	na	na	65	
kSNP raw	100	94	na	
kSNP raw core	na	na	na	
Notes.

na not applicable

Discussion

Although the performance of NGS platforms and the requisite analysis software has been evaluated, those evaluations were often done focusing on a single step in the process (e.g., mapping, de novo assembly, or phylogenetic inference). Here, we have considered the entire process from the sequence platform to phylogenetic clustering. Examination of the full process is important in elucidating best practices to increase the consistency and accuracy of such analyses. We found that within each of the three factors we evaluated there were differences in the results produced and, thus, were able to identify some guidelines for how NGS data should be analyzed for the purpose of clustering.

Variability among NGS platforms

Ideally, different next-generation sequencers would produce identical nucleotide sequences when provided with the same sample. Unsurprisingly however, differences in the chemistry and engineering of different platforms mean they do not agree exactly, but have different error rates and error structure. For example, studies of “benchtop” NGS platforms have repeatedly found that the MiSeq has the lowest single-base error rate followed by Roche’s 454 GS Junior followed by the Ion Torrent PGM (Junemann et al., 2013; Loman et al., 2012; Quail et al., 2012); unlike the MiSeq, homopolymer-associated indel errors were an artifact of the latter two platforms (Loman et al., 2012). SOLiD, with much smaller read lengths than the other platforms investigated, had the lowest sequencing accuracy, lowest coverage rate, and highest false-positive rate when compared to Roche 454 and Illumina GA data (Harismendy et al., 2009).

Given the differences between NGS platforms, we must ask if the differences are significant enough to obscure the relationships among individuals. From our analysis of sequencing runs of the same strain carried out on different platforms and labs, it is clear that replicate runs were not identical (Fig. 3) and, consequently, replicates did not always result in a monophyletic clade (Fig. 4). However, replicates of a strain sequenced on the same platform were more likely to represent such a clade (Fig. 4). And although replicates did not always cluster together they were not likely to be found in different strongly supported clades. These results suggest that it would be ideal to analyze samples sequenced on a single NGS platform but that platform variation is unlikely to result in strongly supported erroneous relationships.

Reference-free vs. reference-based approaches

The analysis of WGS is non-trivial, and there are a number of possible approaches that exist. Here we focused on one possible way that such data could be used: to identify variant sites to construct a SNP matrix from which a phylogeny could be inferred. Some of the approaches (e.g., Mauve; Darling, Mau & Perna, 2010) that can accomplish this are applicable for smaller data sets as they are based on whole genome alignments and were not considered here as we wish to construct a SNP matrix and a resulting phylogeny for 100 s–1000 s of runs. We considered a number of methods that are suitable for large datasets (i.e., reference-free and reference-based) and, therefore, likely represent those most likely to be employed in future studies.

We use a reference-based method in our outbreak investigations because it was assumed to have lower false-discovery rate and naturally excludes variants within the mobilome (e.g., prophages and plasmids), which the results presented here support. However, the primary drawback of a reference-based method is that it will fail to detect SNPs within a region shared among non-reference samples but that is absent in the reference genome either due to using a reference whose genome has not been closed (e.g., an incomplete draft assembly) or the genomic region actually being absent in the reference. In contrast, the reference-free approach will detect SNPs within regions missing from a subset of the runs (depending on the percent presence threshold). This additional detection ability comes at the cost of having to decide what threshold of missing data one should use; we used the 50% majority but other values could include the percent of the dataset that represents the ingroup samples. Without a threshold on the percent of missing data, the reference-free approach may have a much higher false-discovery rate most likely due to sequencing error (e.g., jackpot mutations; (Kraytsberg & Khrapko, 2005)), which is well illustrated by the all matrix using raw reads having approximately three million basepairs. However, even when minimizing the number of false positive SNPs in the matrix through a threshold, the reference-free method still produced more incongruent trees than the matrices constructed with the reference-based methods (Fig. 5).

Phylogenetic inference

The three phylogenetic inference methods we evaluated did not produce the same topology when analyzing the same SNP matrix (Fig. 7). However, many of the topological differences were likely among poorly supported clades (Fig. 5). The topologies produced with RAxML and GARLI did recover the general clustering pattern that differentiates outbreak from non-outbreak isolates more frequently than FastTreeMP. This result is consistent with previous work (Liu, Linder & Warnow, 2011) indicating that RAxML has higher accuracy than FastTreeMP. (This does come with a speed trade-off. For the 194 sample dataset with approximately 50,000 bp per sample analyzed here, a phylogeny could be inferred with FastTreeMP in a matter of minutes and for RAxML and GARLI it was on the order of hours and days, respectively, to obtain a topology and bootstrap replicates.) RAxML and GARLI topologies were fairly congruent with one another for strongly supported branch-points. As a result, for closely related organisms like those studied here there may be a real cost in using faster but less thorough search algorithms.

An important aspect of our phylogenetic analyses is that the substitution models we employed may not be optimal for the types of matrices we were analyzing within which every site is variable. To our knowledge there have been few studies investigating the effects of applying traditional nucleotide substitution models to such matrices. One such study found that under the GTR + Γ model, phylogenetic accuracy decreased when invariant sites were excluded (Bertels et al., 2014). Although these SNP matrices may be better modeled as binary data to which the Mk or Mkv models (Lewis, 2001) could be applied. However, that model is computationally demanding and may not be well suited for the large matrices associated with WGS data; it also has a number of assumptions that SNP matrices likely violate (e.g., an infinite sites model and equal substitution rates between states). Within the RAxML package, the ASC_GTRGAMMA option which models no invariant sites and, thus, may be appropriate but that model also corrects for ascertainment bias, which is not appropriate for the matrices we constructed. Given this potential for model misspecification, an avenue of future research that would strengthen the robustness of inferring phylogenies from SNP matrices would be to determine the consequences of model violations and the development of more appropriate nucleotide substitution models for such matrices

Signal vs. noise

We have evaluated the robustness of topologies to a number of factors that may differ among analyses on a particularly vexing group of samples that are highly clonal and very closely related. Within the original analyses describing the relationships among 47 of the strains, there were only five unique SNP differences between outbreak and non-outbreak isolates (Allard et al., 2012). Within this study the reference-based non-core matrices had six fixed SNP differences while the reference-free approaches had only one. This raises the question of whether there is the possibility that within the current study noise, either due to sequencing errors, variation in SNP detection method, and/or phylogenetic inference, would render impossible our ability to recover the outbreak/non-outbreak structure. Such issues are less likely to arise when investigating a greater degree of evolutionary divergence assuming that a sufficient number of homologous sites can be identified since there will be many more real SNP differences and, thus, a single topology is likely to emerge from different methods. However, due to a number of issues (e.g., lack of a suitable reference genome and difficulties detecting k-mers present in all samples) there will also be an upper bound to the evolutionary breadth of relationships that can be investigated with the methods we have evaluated.

The results presented here indicate it is possible to recover the outbreak structure in a dataset of this type, but care should be taken when deciding on the methods to employ. Methodological choices include: (1) deciding to combine data from different NGS platforms, (2) choosing a method to use for identifying variants, and (3) choosing what phylogenetic inference method should be used. Our results suggest good options for each of these choices and indicate that they allow detection of the biological outbreak signal even among these closely related samples. For example, we found strong bootstrap support for the differentiation of outbreak and non-outbreak samples results when using the non-core matrices constructed using a reference-based method and inferring a phylogeny with either RAxML or GARLI (Table 3). Although we did not evaluate it here, we note that care should also be taken regarding the choice of a reference, as there are a number of studies that have found that topological accuracy and the number of erroneous SNPs detected depends on the reference chosen (Bertels et al., 2014; Pightling, Petronella & Pagotto, 2014).

Conclusions

The robustness of phylogenetic inference to data acquisition and analysis methodology is critically important in many studies (e.g., traceback investigations, systematics, and phylogeography). Our results illustrate how topologies may differ depending on methodological approaches employed—a particularly important issue when investigating closely related samples. Our results also show that good methods do exist that provide robust results when attempting to differentiate between outbreak and non-outbreak samples. For example, the best choice appears to be the use of a reference-based approach allowing for some missing data in the matrix and constructing trees using a thorough inference program. We believe that these general conclusions are likely to be observed across different datasets (e.g., within other taxonomic groups and across a range of genome complexity) than the one investigated here and future studies will reveal whether that assumption is correct.

This work represents our ongoing efforts to build, characterize, and validate our WGS methodology and pipeline for food-borne outbreak investigations. The questions we address are applicable to many in the WGS analysis community, and we hope that the results presented here will contribute to the establishment, through additional investigations, of well-defined pipelines for clustering based on WGS data.

Supplemental Information

Figure S1 Topologies inferred using RAxML for four matrices constructed using the full 194 dataset. Tips are colored according to sequencing platform

Click here for additional data file.

Figure S2 Outbreak color coded tips for the Allard et al. (2012) dataset where phylogenies were inferred using FastTreeMP

Click here for additional data file.

Figure S3 Proportional pairwise distances among individuals based on the different SNP matrices evaluated

The red line is the expected relationship if the two matrices produced the same number of pairwise differences (i.e., x = y).

Click here for additional data file.

Table S1 Sample information

Click here for additional data file.

Supplemental Information 5 16 SNP matrices that were analyzed within this study

See README file for further details.

Click here for additional data file.

We thank Cong Li and Charles Wang for helping to produce whole genome sequence data. We would like to acknowledge Henk den Bakker and Martin Weidman from Cornell University; and Furtado Manohar, Craig Cummings, and Lavorka Degoticija from LifeTech for producing the SOLiD and some of the IonTorrent data. We also acknowledge the staff at the FDA’s Scientific Computing Lab for assistance with running analyses on the HPC. We thank Eric Brown, Peter Evans and Steve Musser for promoting this research.

Additional Information and Declarations

Competing Interests

Author Contributions

The authors declare there are no competing interests, financial, non-financial, professional, personal or otherwise.

James B. Pettengill conceived and designed the experiments, performed the experiments, analyzed the data, wrote the paper, prepared figures and/or tables, reviewed drafts of the paper.

Yan Luo conceived and designed the experiments, performed the experiments.

Steven Davis analyzed the data, contributed reagents/materials/analysis tools.

Yi Chen, Narjol Gonzalez-Escalona and Andrea Ottesen contributed reagents/materials/analysis tools.

Hugh Rand conceived and designed the experiments, analyzed the data, wrote the paper, reviewed drafts of the paper.

Marc W. Allard reviewed drafts of the paper.

Errol Strain conceived and designed the experiments, performed the experiments, analyzed the data, reviewed drafts of the paper.

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
