# Peer review of "An evaluation of alternative methods for constructing phylogenies from whole genome sequence data: a case study with Salmonella"

_PeerJ, doi:10.7717/peerj.620_

## Round 0.1 · original submission · Major Revisions

Both reviewers provided extensive and detailed comments on their concerns with both approaches and conclusions in your study. Both recommend Major Revision. Some of the concerns will take significant effort to accommodate well. Therefore, if you choose to undertake a revision, that revised manuscript would go through another full round of review. However, both reviewers feel the study is timely and covers an important topic. I concur and therefore strongly encourage you to undertake an extensive revision.

·

Basic reporting

This article addresses an important and timely topic, that of applying DNA sequencing and phylogenetic inference to characterize ongoing outbreaks of bacterial pathogens. In general the writing is clear although I felt that the structure of the article could be improved by shifting the methods section to precede the results. This is a methods-focused article and understanding of the results is simply not possible without having first read the methods (although some of the more technical details in the methods are not required to understand results). Several of the figures would benefit from improvement in order to communicate information more effectively and I have made some specific suggestions about that below.

Experimental design

An attempt has been made to evaluate several different strategies for the sequencing and phylogenetic analysis of bacterial genomes. The authors have collected a very nice dataset where the same isolates have been sequenced on different high throughput sequencing platforms, then analyzed using a variety of different approaches. Unfortunately there is a major weakness with this approach to comparing sequencing platforms, which is that the true evolutionary history is unknown, so it is impossible to know how accurately any particular phylogenetic reconstruction reflects the truth. The authors acknowledge this issue and instead focus on comparing approaches to see which produces the most consistent results from run to run. This is measuring variance in the methods and not accuracy and it is absolutely critical that the article be revised to make the distinction clear.

Part of the analysis focuses on the ability of various approaches to distinguish a collection of outbreak-associated isolates from other isolates. There is an implicit assumption that the outbreak strains are truly monophyletic. I am not very familiar with that particular dataset and it would help readers like me if the basis for this assumption were explained, for example if there are additional lines of evidence apart from sequence data that support the assertion such as spatial or temporal data. In particular it would be helpful to understand why it's impossible for a non-outbreak isolate in the dataset to share ancestry with the outbreak strains.

Another major issue that must be addressed more directly in the manuscript is how uncertainty in the phylogenetic reconstructions was dealt with. When an inference program produces a topology that conflicts with the expected topology are incompatible splits included with high support values? gsi apparently has the ability to account for uncertainty by processing bootstrap replicates, and it may be possible to employ a similar approach with the other topological distance metrics.

Validity of the findings

It is not clear that all of the claims made in the discussion and conclusion section are supported by the data. In particular, more information is required on how support values (whether they are SH-like or bootstrap supports or something else) relate to the apparent topological discordance. I have highlighted a few specific places where more data is required to substantiate claims in my comments below.

Additional comments

I have a number of specific comments that I have listed below.

Fig 2C, log scale would help --
Fig 2, This plot is misleading because it suggests that the HiSeq variation is very low, but this is just because there is a single sample while the other platforms have many! The data needs to be presented in a way that communicates both number of samples and the variance in the distribution.

Why bother including the HiSeq data at all? With only a single sample there is essentially no statistical power to say anything meaningful about how this platform compares to the others.


Line 104, what metric is being used to calculate the pairwise differences presented in Figure 3? I think it is described later in methods but it would help to mention where to find the metric.

Line 105, what about replicate runs on the HiSeq?

From Fig 3 it looks like SolSNP and bcftools might be calling the same differences because their patterns are similar, but it is difficult to know with any certainty. It would be interesting to know if the variation among replicates is driven predominantly by variation in the sequence data rather than the algorithm. one way to assess this would be to take all the pairwise differences between all runs (e.g. bcftools vs. bcftools, SolSNP vs. bcftools, SolSNP vs. SolSNP, not sure if kSNP can be included) and apply some type of dimensionality reduction (MDS, PCA, etc) to determine if most of the variation corresponds to differences among replicates rather than differences among SNP callers

Line 122: "Our results indicate that a matrix that accepts some level of missing data, while not including all
possibilities is optimal for phylogenetic inference" this is a conclusion, but the data which leads us to this conclusion has not yet been presented!

Line 129: "Although some of these additional SNPs may be due to
mobile elements, it is also likely that many of them are erroneous, which may explain the
decrease in bootstrap support despite the larger matrix when compared to the de novo majority
matrix" I don't understand what leads the authors to this conclusion. What evidence exists that these are erroneous rather than a product of recombination among strains? The loss of bootstrap support values when the matrix includes increasing levels of missing data is a common phenomenon I believe, and derives from the fact that bootstrap sampling of a matrix with a high fraction of missing values tends to include few of the sites with complete information, causing tree inference on that replicate to be poorly informed.

Figure 6: Many of the nodes overlap each other and this often obscures high or low confidence nodes. This makes it hard to read the figure. One way to resolve the issue would be to use alpha channel blending (e.g. translucency) when plotting the cladogram.

Figure 7 & 8: what are the branch lengths? if possible their meaning should be described. Also, it looks like the topologies sorted according to some branching order metric, however this causes the tanglegrams to look a lot more tangled than they might be if the topologies were sorted to minimize crossing lines in the tanglegram. It makes the degree of topological discordance very difficult to discern in the figure.

Fig 9: in its current form this figure is not as helpful as it could be. There appear to be subtle differences among the trees but this is really hard to see when the collection of 8 divergent strains are included.

Line 155: "...we are working under the assumption that methods that result in similar topologies are optimal."
It is good to see this acknowledged explicitly because it certainly seems to be implied elsewhere in the presentation of the material. Still, this is faulty logic. My buddy Marc Suchard and I could write two different tree inference programs that give the same tree every time, that doesn't make the tree right. The fact that two likelihood based inference methods employing very similar models of evolution (GARLI and RAxML) yield similar trees might just mean that they methods have the same inference bias.

Line 333: I am curious about the impact of single-end versus paired-end read mapping, although any change in results seems likely to be dwarfed by the apparent run-to-run variation

Line 345: "if different nucleotides were called at the position, the one with frequency larger than 50% was the consensus call for that position" 50% support seems a bit low when there is clear ambiguity in the base call data that will almost certainly influence the phylogenetic inference in some way. it seems like a higher threshold would improve precision in the phylogeny, albeit with a minor loss of resolution

Line 350: "www.github.com/XXXX" I will not be able to review the code until the correct link is provided

Line 400: how big are these groups? Seems like they should all be pretty small. How does the gsi statistic behave with small groups of e.g. size 2?

Line 416: is there any particular reason for using the bootstrapping via seqboot instead of the SH-like support values that FastTree calculates by default?

Line 427: how were clade confidence estimates done with raxml? it supports at least a few different approaches.

Line 152: the claim is made that "many more topological rearrangements are required to reconcile FastTreeMP tress with either GARLI or RAxML" and the reader is directed to Figure 8. It is very very hard to see from this figure how many topological rearrangements are actually required to transform one topology into another. A small number of changes deeper in the tree can create a large number of crossing lines. It would be much easier to interpret a quantitative metric of topological distance such as the symmetric difference, SPR distance, or the generalized robinson-foulds. A second issue relates to the degree of support. It sounds like, based on the methods, that fully resolved trees are being evaluated (at least for GARLI, this info was not given for raxml or FastTree) instead of e.g. 50% or 90% majority rule consensus trees. When there is a great degree of uncertainty in the data, such as would be the case with a large collection of closely related strains, this means that an arbitrary pair of equally likely fully resolved trees may have large topological distances.

Line 224: why not mention the case where a closed, finished reference is actually missing a gene common to most members of the species?

Line 240 and 175: I do not understand how the conclusion was reached that FastTreeMP incorrectly grouped the outbreak isolates with non-outbreak isolates. Looking at supplemental fig S2 which contains the FastTree results, many of the panels appear to have placed the outbreak strains into a monophyletic group (de novo all, de novo majority) or very nearly so with only one error (snpmat, solsnp, raw majority). More importantly, in cases where FastTree has incorrectly grouped strains, has it placed a high support value on that split? If not then data does not support the assertion that FastTree is doing any worse than the other methods.

Line 248: this assertion is only supported if the statement on line 240 is supported.

Lines 260-262: agree that more work is needed on model selection for genome SNP data!

Line 267: it would be helpful to mention how many SNPs have a pattern that is incongruent with the outbreak vs. non-outbreak split

Line 294-296: whether these conclusions are valid depends on how the FastTreeMP support values look



The genealogical sorting index (gsi) is a nice way to compare groups in trees, but several important details of how it was applied here seem to be missing. gsi normally requires rooted trees, I believe. How were the trees rooted? Also, gsi can be applied to the entire collection of trees in a bootstrap set, thereby enabling some of the uncertainty in the inference to be captured in the score. Was that done here? Given the great deal of uncertainty present when dealing with such closely related strains it almost certainly should.

·

Basic reporting

No comments

Experimental design

The authors refer to a Github repository with Python scripts to reproduce the work, however the URL is not complete (www.github.com/XXXX). Please correct this before publication.

The SNP filtering technique used (AF1=1, >=10x coverage) is probably not the best. The Samtools base-caller expects diploid data and attempts to fit the data to a model of allelele frequencies of either 0, 0.5 or 1.0. This is probably not a good choice for bacteria. We prefer to use something like VarScan2 and use a specific alelle frequency cut-off (e.g. >90%). Other groups do something similar. Insisting on >10x coverage may penalise the 454 data.

I would have liked to have seen a basic recombination filter e.g. >3 SNPs in 1000 as is standard practice employed too.

I am not clear the justification for using GTR+CAT with FastTree and whether any other models were tried.

Validity of the findings

I think the authors are to be commended for tackling a difficult subject, and I suppose I agree with many of the conclusions in general. I think the results showing that the k-based methods are inferior to reference-based methods are important.

However it is probably worth stressing that these results may not be reproducible between different versions of sequencers and different protocols for aligning, SNP filtering and phylogenetic inference.

There is a bit too much emphasis placed on the differences in tanglegrams when the authors conclude that these changes tend to occur in branches without strong support. It is a shame a better tree could not have been constructed (would recombination have helped?).

I am _really_ surprised about the differences between RAXML, GARLI and FastTree and I am wondering if anything has gone wrong here, and wonder if this is something to do with the SNP matrix construction. This does not fit with my experience, or with e.g. Liu 2001. Possible ideas are a lack of recombination filtering, some error with the scripts or the wrong choice of substitution model. Given the difference in runtimes between such programs, I feel this should be bottomed-out before publication to check nothing funny is going on.

I would like to see the data (e.g. VCFs, SNP matrices) to comment further. It would be good if the Github repository could be populated with the relevant scripts and data before resubmission.

Additional comments

This is a technical article but a little more time on whether any of this actually matters, and how it might matter in an outbreak situation would have been useful. I doubt, for example, that these differences in phylogenetic reconstruction would affect conclusions during outbreak management, for example. Can you comment?

---

## Round 0.2 · Minor Revisions

Thank you for your effort in taking into account the previous reviews and updating your paper accordingly. I had a previous reviewer look it over and s/he was very happy with your revisions. However, a new reviewer has identified a few issues that still need to be addressed. I think your paper will be improved by taking into account these additional minor edits. Otherwise, you are nearly there!

·

Basic reporting

No Comments

Experimental design

No Comments

Validity of the findings

No Comments

Additional comments

I would like to thank the authors for their considered responses to my initial review. I think this manuscript is now much improved and will be a valuable contribution to the literature. I think this may help spark a useful discussion into considering standards of reproducibility and reporting in epidemiology studies employing whole-genome sequencing.

·

Basic reporting

The paper is well written and generally very clear. There were a few typos throughout, just requires one more proof-read I think.

1. The use of ‘strain’ throughout is problematic; it would be more appropriate to replace ‘strain’ with ‘isolate’.

2. Prior literature is appropriately introduced and referenced in the background section. A few comments:
- Intro, last para -> clarify whether “replicate runs of the same strain” refers to sequencing of the same library, same DNA extract, or fresh extract + library prep on the same isolate; if fresh extracts, were they from the same subculture? It seems from the results that there are a mix of scenarios.
- line 136 -> “we do have replication of a number of strains and, therefore, have expectations for how those isolatesshould cluster (i.e., be monophyletic) and the number of SNP differences that should exist among them (i.e., zero).” The latter (zero SNP differences) is true if sequencing the same DNA extract or library; not necessarily if sequencing DNA extracted from different subcultures of the isolate.

3. Some minor changes would improve readability of Figures and Tables:

Figure 2
-> please indicate the sample size for each platform
-> for HiSeq, there is only one sample, which is inappropriate for a boxplot; the value for this sample should be indicated with a single point

Figure 4
-> please indicate the number of data points (i.e. resequenced sample pairs) contributing to each box

Figures 5, 7
-> please indicate how the violin plots should be read (e.g. as was described for box plots)
-> why switch from box plots to violin plots at this point in the paper?

Figure 6
-> the circles are mostly overlapping and therefore difficult to see the colours. It may be clearer to colour the branches by their support values rather than the nodes?
-> It may be clearer to show these as circular dendrograms, as in Figure 8.
-> There appears to be grey background shading, however I think this may just be the product of very closely spaced lines? Again, this would be avoided if the trees were shown as circular dendrograms as in Fig 8.

Figure 8
-> which phylogenetic method was used to infer these trees? e.g. in Fig 6 this was RAxML

Table 2
-> replace ‘bcftools’ with ‘VarScan’?

Table 3
-> what does ‘BP’ mean? I think this could be removed as the legend states the table is indicating bootstrap support?
-> Does ‘NA’ mean that this bifurcation did not occur in the tree? i.e. that the outbreak sequences did not form a monophyletic group within the tree? Please clarify in the legend.

Experimental design

1. Some details are missing from the Methods section:
line 158 -> What parameter settings were used in the bowtie2 mapping? What impact might these choices have?
line 162 -> SamTools was used to generate a pileup. What parameters (e.g. mapping qualities) were used? What impact might these choices have?

2. lines 171-176 “First, a ‘snplist’ file is generated by identifying all variant sites across the individual .vcf files. Second, for each sample an additional pileup file is generated based on only the positions contained in the ‘snplist’ file. This second step primarily determines the nucleotide state for positions within a sample that were not present in the .vcf file.”
-> I think the authors mean that the ‘snplist’ file contains all those variant sites that were identified among a set of .vcfs that represent the full set of isolates to be compared? Could this be phrased more clearly.

3. lines 195-199. The authors use two SNP matrices - One with all SNP sites regardless of missing data; and one with no missing data (the ‘core’ site alignment). Neither of these scenarios seems ideal; could they also include a matrix that contains ‘common’ sites, e.g. those with <5% or <10% missing data across samples?

4. Phylogenetic inference:
Three software packages were used for phylogenetic inference - these programs all use maximum likelihood (ML) to estimate the topology and branch lengths.
- What was the rationale here? Why not consider e.g. Bayesian inference, or a distance-based tree?
- lines 263-282: The models used with each of the programs are slightly different. What is the rationale here? What are the authors aiming to assess here?
(i) Why was the GTR model chosen? Was this the best fit to the Montevideo data using ModelTest or similar?
(ii) How were the specific models chosen for each program? Why were they different? Is this a reasonable comparison? E.g. FastTreeMP used GTR CAT with 4 categories; GARLI used GTR+I+Gamma; RAxML used GTR CAT with the default (presumably) 25 categories… why not use the same model for all?

Validity of the findings

1. Platform reproducibility, text (lines 306-317) & Figure 3: “To characterize platform variability, we examined replicate runs of the same sample sequenced on the same platform (Fig. 3). […] We note that many of the replicate samples are not pure platform replicates, but were from independent cultures often performed at different labs, which means our observed differences are not just due to platform variability.”
-> The conclusion drawn here is that MiSeq produces more reproducible data than the other platforms. However from the data shown, one cannot tell whether this is confounded by other sources of variability. The authors state the data includes different kinds of ‘replicates’ - they should show how this is distributed amongst the 3 platforms so that one can assess whether this is impacting on the platform comparison. E.g. were the MiSeq ‘replicates’ mostly replicate MiSeq runs of the same DNA libraries? Or replicate libraries on the same DNA extracts? Or replicate DNA extracts on the same culture in the same lab? Or do the MiSeq replicates include different analyses of subcultures of the same isolate sent to different labs? An ANOVA may help to assess whether the importance of these sources of variability compared to platform.

2. line 350-355:
“In conclusion,
351 within a reference-free approach, it is probably optimal to use the actual
352 number of SNPs detected rather than cull positions with missing data; for a
353 reference-free approach the threshold of missing data that results in optimal
354 information content is more complicated and depends, in part, on the
355 evolutionary breadth of the samples being investigated.”
-> Do the authors mean “within a reference-BASED approach, it is probably optimal to use the actual number of SNPs detected”?
-> What does ‘use the actual number of SNPs detected’ mean? I think the intended meaning is to ‘use all SNP sites detected’, but ‘use the actual number of SNPs’ could be taken to mean using SNP counts / distances for clustering.
-> As mentioned earlier, why not consider a matrix that contains ‘common sites’, e.g. <5% missing data, rather than taking an all or zero approach to missing data?

3. Phylogenetic inference (lines 370-374)
“there was greater topological congruence when comparing a tree produced under RAxML and GARLI than when either of the trees produced under those methods were compared to one inferred with FastTreeMP. […] As we do not know the true topology, we note that when comparing the different inference methods we are working under the assumption that methods that result in similar topologies are optimal.”
-> The flaw in this rationale is that GARLI and RAxML implement very similar algorithms, whereas FastTreeMP uses an approximation. Therefore one would expect GARLI and RAxML to give more similar results to one another than to FastTreeMP, regardless of the input alignment; i.e. their agreement is not a good measure of whether the topologies inferred are in some sense ‘optimal’, it is simply that they are both using the same criteria to decide between alternative topologies. This rationale would be more valid if the methods under comparison used independent (non-ML) approaches to phylogenetic inference, like Bayesian inference (e.g. BEAST or MrBayes) or simple distance-based methods.

4. I would like to see the authors comment on the generalisability of their findings. S. enterica Montevideo is a highly clonal bacterium of ~4.5 Mbp and ~50% G+C content. But bacterial pathogens differ widely in their population structure and genomic properties - how generalisable are these findings likely to be to other bacteria with smaller genomes? high or low G+C content? faster mutation rates? higher recombination rates? The authors make no mention of recombination and its potential impacts on phylogenetic inference. This is probably because recombination is not a major factor in the microevolution of S. enterica serotypes, but this issue should at least be mentioned in the Discussion, given that recombination is a major factor in many other bacteria that are now being regularly investigated with WGS.

Additional comments

1. lines 404-407 -> 6 SNPs differentiated the outbreak strains in reference-based but not reference-free analyses; can the authors expand on this? Where were these SNPs located? Are they in repetitive regions? Can they explain why these loci were not detected in reference-free analysis?

2. In the Discussion:
“An important aspect of our phylogenetic analyses is that the substitution
494 models we employed may not be optimal for the types of matrices we were
495 analyzing within which every site is variable. To our knowledge there have
496 been no studies investigating the effects of applying traditional nucleotide
497 substitution models to such matrices.”

This issue was addressed in a paper published in March this year by Bertels et al (http://mbe.oxfordjournals.org/content/31/5/1077.long) who used simulated data to show that phylogenies based on SNP sites (i.e. excluding invariant sites) were less reliable than those that include invariant sites. They also showed that the choice of reference sequence could impact the inferred tree topology and suggest a method (and software) for generating alignments of variant + invariant sites based on multiple references. These points are highly relevant to the present manuscript and should be cited in the Discussion.

---

## Round 0.3 · accepted · Accept

Thanks again for your hard and thorough work responding to the reviewers' constructive critiques.